# Method for Operating Drainage Pump Stations Considering Downstream Water Level and Reduction in Urban River Flooding

Yeon-Moon Choo [1] , Jong-Gu Kim [2], Shang-Ho Park [2], Tai-Ho Choo [2] and Yeon-Woong Choe [2,*]

1 Institute of Industrial Technology, Pusan National University, Busan 46241, Korea; chooyean@naver.com
2 Department of Civil and Environmental Engineering, Pusan National University, Busan 46241, Korea; koreaws@empas.com (J.-G.K.); sangogo@nate.com (S.-H.P.); thchoo@pusan.ac.kr (T.-H.C.)
* Correspondence: ywchoe@pusan.ac.kr; Tel.: +82-051-510-7654

**Abstract:** Korea experiences increasing annual torrential rains owing to climate change and river flooding. The government is expanding a new drainage pump station to minimize flood damage, but the river level has not been adjusted because of torrential rains. Therefore, the river level must be adjusted to operate the drainage pump station, and it can be adjusted through the reservoir of the drainage pump station. In this study, we developed a method for operating drainage pump stations to control the river level and verify the effectiveness of the proposed method. A stormwater management model (SWMM) was used to simulate the Suyeong River and Oncheon River in Busan, Korea. The rainfall data from 2011 to 2021 were investigated. The data were sorted into ten big floods that occurred in Busan. The model was calibrated with actual rainfall data. The water level of the Suyeong River and the Oncheon River was the highest in most simulations. The simulation results showed an average decrease of 3018.2 m$^3$ in Suyeong River flooding, and the Oncheon River needed to be supplemented due to structural problems. As a result of the recombination by simply supplementing the structural problems of the Oncheon River, the average flooding of 194.5 m$^3$ was reduced. The proposed method is economical and efficient for reducing urban stream flooding in areas susceptible to severe damage caused by climate change.

**Keywords:** urban inundation; drainage pump station; flood reduction; SWMM

## 1. Introduction

Urban streams are flooded annually because of climate change, and the resulting damage is severe [1–4]. In particular, coastal cities such as Busan, Korea, experience flood flowing from the upper reaches, and the tidal level influences the damage [5]. Torrential rains fell in Busan in 2014 and 2020, and the high tidal levels caused significant damage. In the 2014 events, seven people were killed and Kori Nuclear Power Plant 2 in Busan was shut down. In 2020, 45 people were killed, and there were approximately 7000 victims in South Korea. During summer in Korea, there was a rainy season in which approximately 300 mm of rain fell for 30 days. However, more than 200 mm of torrential rain each day does not fall frequently; hence, there are no policies to prepare for torrential rains. Therefore, there is a need to prepare for torrential rains because of climate change in tropical regions. The main streams in Busan are the Suyeong and Oncheon Rivers. The Suyeong River has a river extension of 26.4 km and a catchment area of 200.06 km$^2$, and it is discharged by Suyeong Bay [6]. As most watersheds pass through the city center, flooded rivers cause severe damage. Because approximately 570,000 people live in the lower reaches of the Suyeong River and most vehicles are parked in underground parking lots due to the high population density, human casualties and property damage are high when the river is flooded. The Oncheon River has a river extension of 12.24 km and a catchment area of 56.28 km$^2$, and it flows into the Suyeong River. Most of its catchments run through the city center and are old. In addition, the Oncheon River has an impermeable area of approximately 49%, which surrounds most of the river. Therefore, the Oncheon River is prone to torrential

rainfall. It is typically flooded every year because of its average slope of 12.53°, its low level, and relatively narrow average descent of approximately 49 m compared to the Suyeong River [7]. Figure 1 shows a satellite map of the Suyeong River basin downstream and the entire Oncheon River basin. From Figure 1, the downstream of the basin consists mostly of urban areas. The land use ratio of Suyeong River and Oncheon River is shown in Table 1.

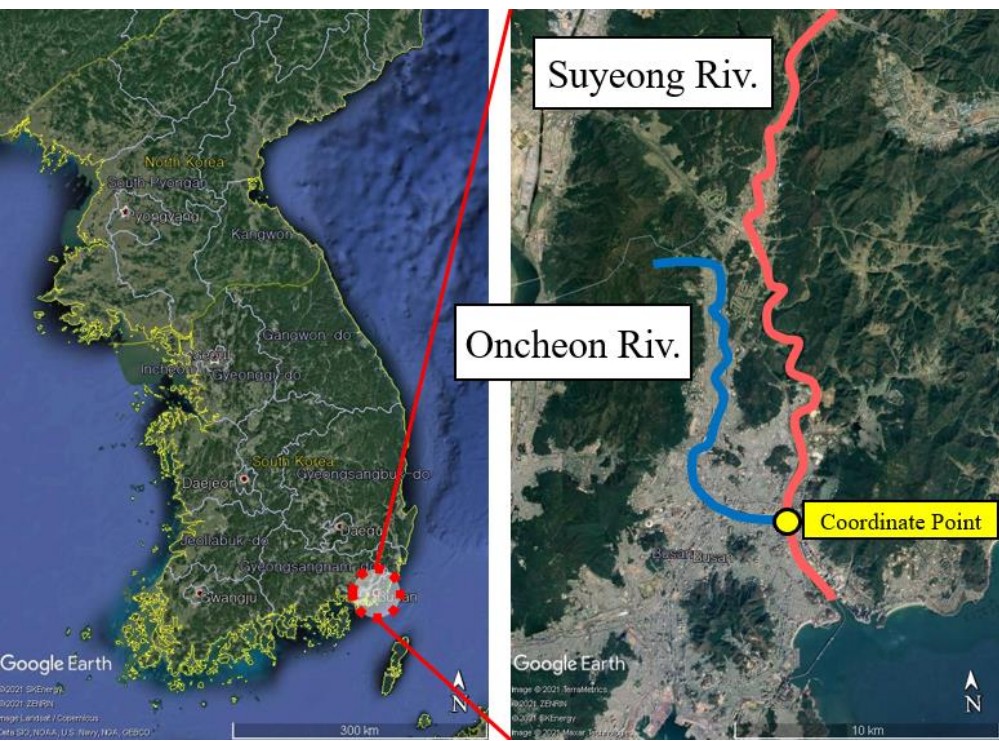

**Figure 1.** Study area (Coordinates: 35.18892898003925, 129.11526723161828).

**Table 1.** Land use and cover structures in the study river basin (using GIS data).

| Sortation | | Sum | Crop Land | Forest | Urban Areas | Water Area |
|---|---|---|---|---|---|---|
| Suyeong River | Area (km²) | 200.06 | 41.68 | 88.84 | 45.66 | 23.88 |
| | Ratio (%) | 100.00 | 20.83 | 44.41 | 22.82 | 11.94 |
| Oncheon River | Area (km²) | 56.28 | 10.43 | 7.12 | 27.73 | 11.00 |
| | Ratio (%) | 100.00 | 18.53 | 12.65 | 49.27 | 19.55 |

Many cities worldwide, including Busan City, developed around rivers. Therefore, different techniques for preventing river flooding are investigated recurrently. Winsemius et al. developed a method for predicting flood risk globally to prevent river flooding [8]. Kundzewicz et al. analyzed historical flood information to prevent the continuous flooding of European rivers, although flood risk mitigation involves high costs [9]. Furthermore, structural (such as dams and reservoirs) and nonstructural methods (such as flood prediction and warning systems) have been proposed to minimize flood risk in China [10]. Because China has a very long history, the technology to prevent large rivers from flooding has been existed for a long time. However, due to climate change and as the city grows, structural solutions have been limited. So, like Kundzewicz et al.'s thesis, nonstructural flood prevention measures are actively being studied in China. Currently, in Europe, including the Czech Republic and Russia, related research is studied, and this is a global trend. Since this paper also proposed a nonstructural flood reducing method, it might help countries with similar problems. Duií et al. conducted innovative research in the Czech Republic to determine approaches for reducing flood risk through households

rather than facility construction or government policies [11]. Choo et al. established a real-time flood prediction method using a flood nomograph different from conventional flood prediction methods [12].

Dams have been installed in existing watersheds to prevent river flooding. Blazkova et al. applied a method to estimate the flooding frequency of dams in large catchment areas in the Czech Republic [13]. Badenko et al. conducted a study to transform existing dams around the Russian Far East into multipurpose dams [14]. Recently, Choo et al. analyzed flood reduction rates in downstream watersheds to convert existing water supply dams into multipurpose ones [15].

Furthermore, studies have been conducted to reduce flooding using rainwater storage tanks and drainage pump stations. Harimurti developed a method of utilizing rainwater storage tanks and drainage pump stations to control flood peaks and flow rates in flood-prone areas in Indonesia [16]. Kim et al. analyzed the influence of a stormwater management model using an underground parking lot as a rainwater storage tank to reduce urban flooding [17]. Jafari et al. optimized the operational performance of gates and pumps using a stormwater management model to reduce urban flooding risk and peak water levels in pumping stations [18]. Chen et al. reported that downstream drain management is crucial in flood management because if the capacity of the pump station located downstream is insufficient, new flooding hotspots are created downstream, and the risk of downstream flooding increases [19].

The types of pump stations include drainage pump stations and wastewater pump stations. The drainage pump station is a pump station installed to drain rainwater directly into nearby rivers. The wastewater pump station is a pump station installed to pump water to a drainage pump station or sewage treatment plant by pumping water to the surface. The depth of the burial deepens as the water goes down through a long pipeline. In this paper, we conducted a study to reduce flooding in areas where rivers are frequently flooded and where drainage pump stations are installed near rivers.

New disaster prevention facilities have been built in Busan City, but incidents continue to occur. In Busan City, 15 drainage pump stations are currently located downstream of the Suyeong and Oncheon Rivers, where flooding occurs frequently. The current drainage pump stations are operated using a water gauge installed at each station, but the river level is neglected [20]. If the rise in the water level of the river is ignored, the possibility of river flooding increases and many catchments where rainwater drains into the river will be flooded because of slow draining.

In addition, Busan City currently operates drainage pump stations separately for each drainage pump station. In emergency cases, a "control tower" constructed in Busan City Hall is monitored by humans and is used to direct the drainage pump stations. Torrential rains fall suddenly and are difficult to manage because they require extended time to form a "control tower".

Therefore, the existing method of drainage pump station operation should be modified to reflect torrential rainfall, which occurs increasingly often due to climate change. In this study, a new method of operating drainage pump stations is developed to take the influence of climate change into consideration. The method can be shared with researchers around the world who study nonstructural flood reduction measures. This study reduces flood through new ways of operating by of utilizing existing methods.

## 2. Methods

### 2.1. Outline

In this study, a new method for operating a drainage pump station was developed and compared with the existing operation method. A stormwater management model (SWMM) was established for analysis [21]. SWMM is a hydrological analysis program which can be used for flood simulation. Furthermore, the control function allows coding of the model, which allows for commanding different operating methods for different

facilities and conduits. The algorithm for operating the drainage pump station was coded in the SWMM. The overall research flow is shown in Figure 2.

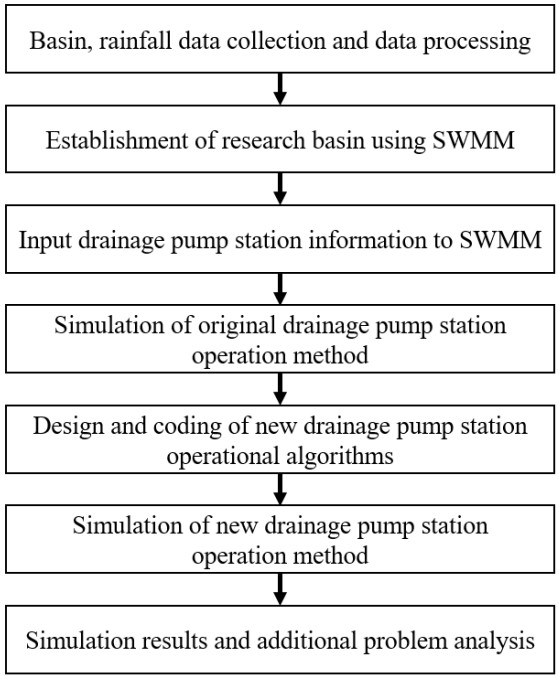

**Figure 2.** Research Flow.

### 2.2. Rainfall Data

Rainfall data recorded for Busan within the past 10 years was collected. Historical rainfall data were collected using the automatic weather station (AWS) provided by the Korea Metropolitan Administration [22]. AWS is a ground observational station managed by the government, and it records various meteorological data, such as temperature, precipitation, wind, humidity, and atmospheric pressure. The collected data were daily data obtained per minute from 2011 to 2021. Data with rainfall values close to 200 mm were used (Figure 3).

### 2.3. Research Basin Composition

In this study, the entire Suyeong and Oncheon Rivers in Busan were modeled using the SWMM. The main parameters used in this model are shown in Table 2.

The river model was established using drawings measured in 2014 (Suyeong River) and 2017 (Oncheon River) and provided by the Busan Metropolitan Government [6,7]. In addition, catchment information such as catchment slope and impermeable area, rainwater pipe networks, and drainage pump stations in the catchment areas were inputted based on the most recent data provided from the Information Disclosure Port and Public Data Portal of the Korean government [23,24]. The last point of the Suyeong River drains into the ocean (Suyeong bay). Therefore, a tidal curve must be entered at the last point of the Suyeong River. Figure 4 is the average tidal curve of Suyeong bay entered into the model.

Five of the 15 drainage pump stations within the Suyeong River and Oncheon River basins were constructed. The selection was based on whether there was a reservoir at the first drainage pump station. The second criterion was whether there was a river downstream. These criteria were adopted because a reservoir should be available to store rainwater until the river level reduces, and downstream areas are susceptible to frequent flooding. Thus, the flooding should be simulated to reduce the water level at the downstream areas. Figure 5 shows the locations of the built drainage pump stations and the area under frequent flooding. Table 3 lists the data for the constructed drainage pump stations. Each drainage pump station has a reservoir for storing water. The reservoir and

the drainage amount in the drainage pump station are actual dimensions and are entered into the model.

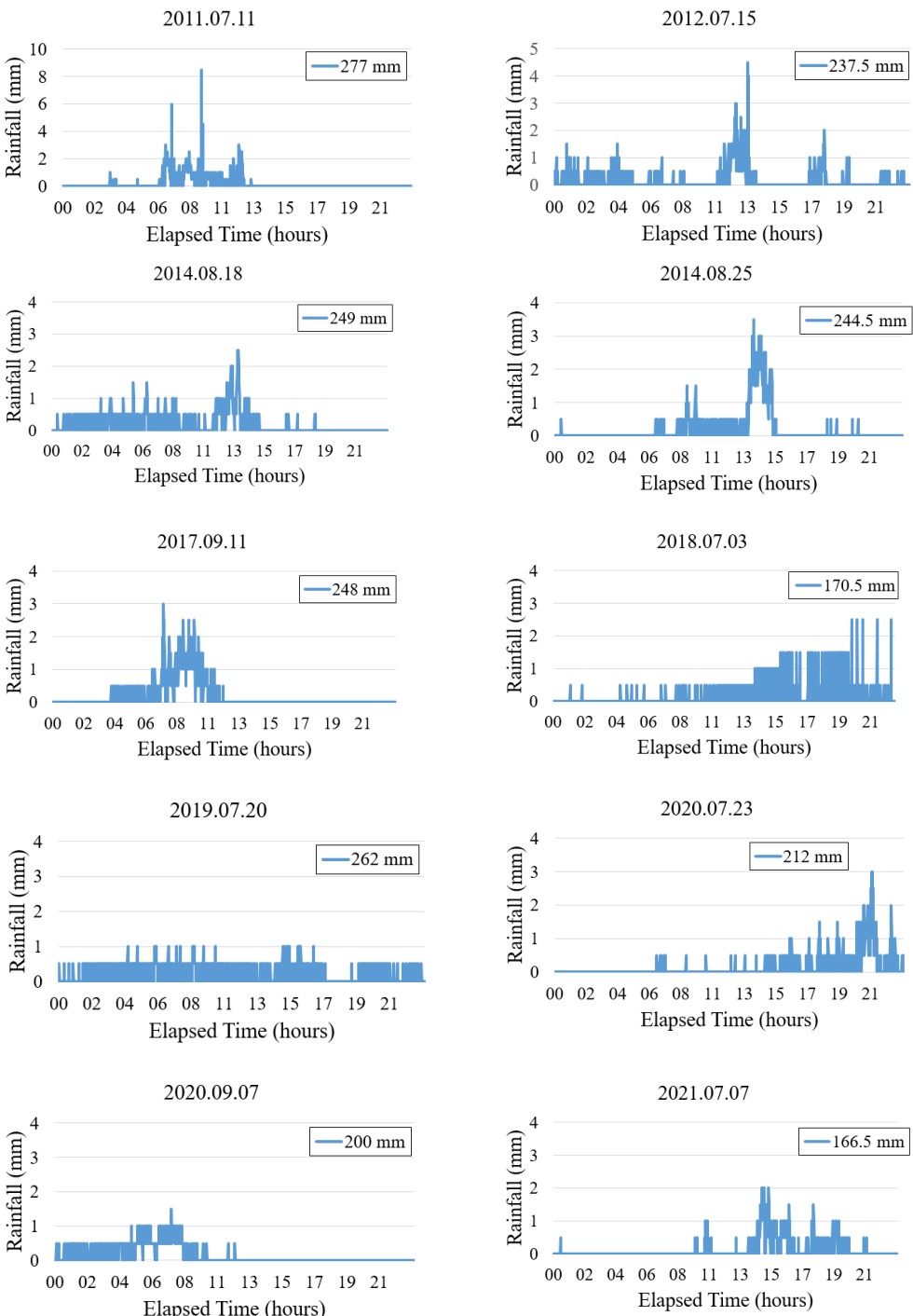

**Figure 3.** Rainfall graphs.

**Table 2.** Main Input Parameters of SWMM.

| | Input Parameters | | |
|---|---|---|---|
| Node | · Inveer El.<br>· Max. Depth<br>· Initial Depth | · Tidal Curve<br>· Storage Curve | |
| Links | · Max. Depth<br>· Length<br>· Roughness | · Inlet Offset<br>· Outlet Offset<br>· Initial Flow | · Pump Curve |
| Subcatchments | · Area<br>· Width<br>· Slope (%) | · Impervious Area (%) | |

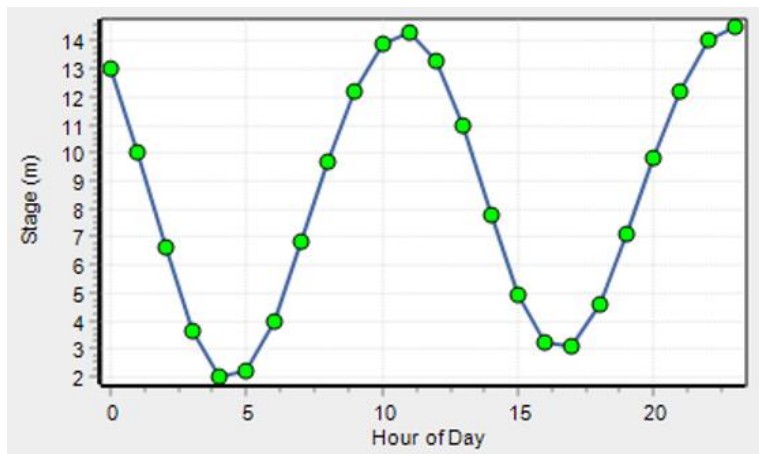

**Figure 4.** Average tidal curve of Suyeong bay entered in SWMM.

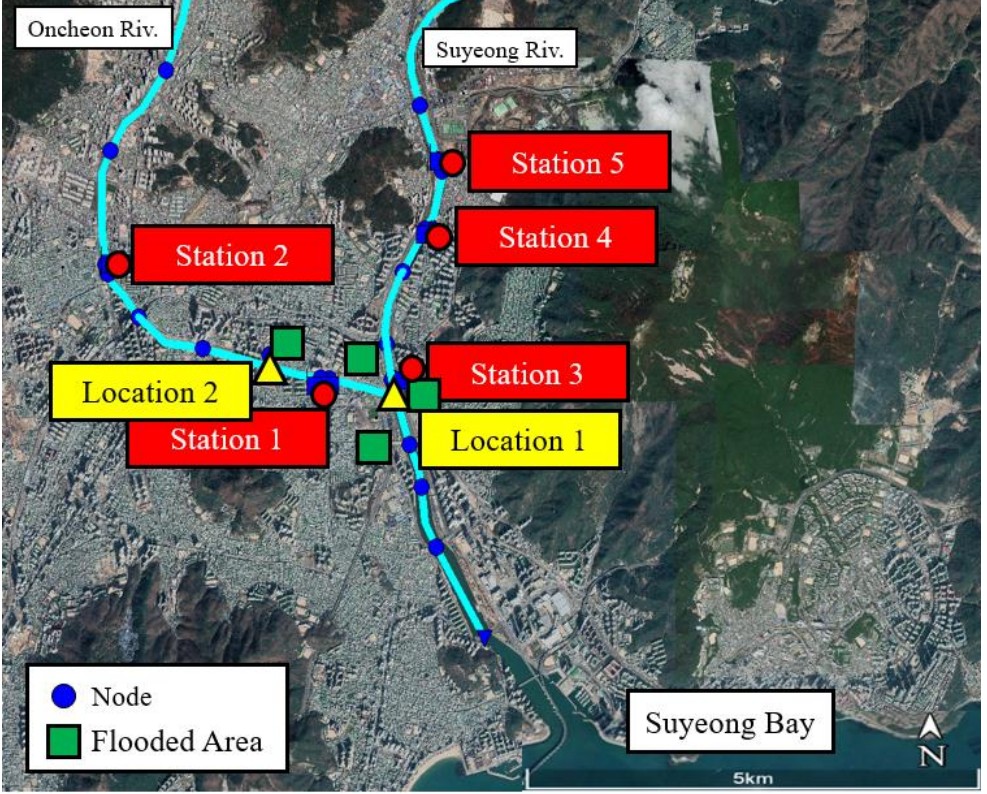

**Figure 5.** Locations of drainage pump stations based on flood analysis point (in SWMM).

**Table 3.** Drainage pump station data.

| Parameter | Station 1 | Station 2 | Station 3 | Station 4 | Station 5 |
|---|---|---|---|---|---|
| Reservoir area (m$^2$) | 10 | 400 | 500 | 38 | 1010 |
| Reservoir capacity (m$^3$) | 61 | 1200 | 1700 | 75 | 8500 |
| Drainage amount (m$^3$/min) | 100 | 784 | 100 | 44 | 1460 |

The flood point shown in Figure 5 represents a collection of possible damage cases from 2014 to 2020 after sorting out the damage caused by river flooding and then combining the areas where damage occurred more than five times in the same area. Most of the damage occurred at the junction between the Suyeong and Oncheon Rivers, and a significant amount occurred near Station 1. Thus, the simulation results for Locations 1 and 2 were analyzed. There are 50 nodes, 48 conduits, and five drainage pump stations that we entered into the model.

*2.4. Algorithm Construction*

The proposed method does not reflect the current water level. Nevertheless, when the stream water level increases to a dangerous degree, the drainage pump station with a reservoir stops operating and then resumes after the flow has escaped (Figure 6).

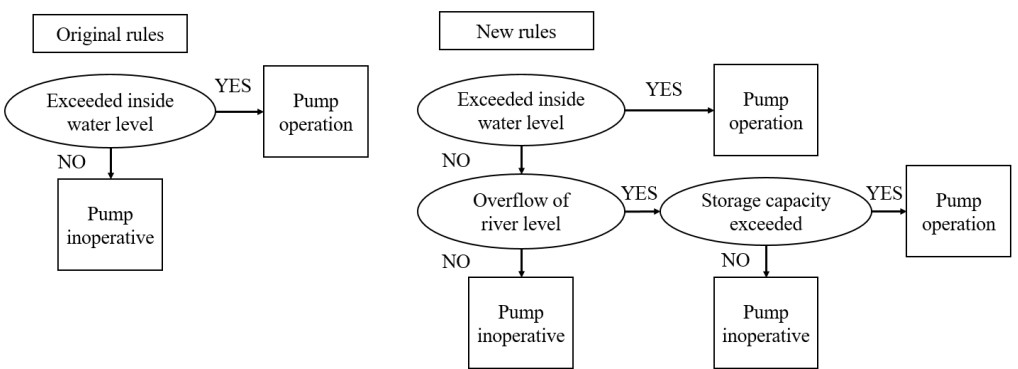

**Figure 6.** Original and new rules of the drainage pump station.

Algorithms such as "new rules" in Figure 6 were coded using the control function in SWMM, and the existing and new drainage pump station operations were compared.

For algorithm optimization, we have designed the system to be as simple as possible based on the original rules. The reason is to control flooding of rivers while achieving the purpose of drainage pump stations. Therefore, the first goal of the new rules is to reduce inland flooding and the second goal is to turn on/off pumps according to the flood level of the river. The constraint of this algorithm is the inland water level. If the inland water level is not stable, the drainage pump station will operate even if the river is flooded.

Furthermore, the limitation of this algorithm is that SWMM provides only limited coding methods. Therefore, this algorithm cannot simultaneously produce a failure situation of a drainage pump station or a situation where manual operation is required due to overload of the drainage pump station. The solution is to prepare and simulate two algorithms: this algorithm and an additional situation algorithm. The results are then tracked through statistical analysis. This solution can be applied reflecting the characteristics of the local drainage pump stations and the government policies of the region that wants to apply for future study.

The "new rules" are more complicated than the "original rules", but they do not require the replacement of existing pumps; they only require the programming of the current drainage pump station system to operate similar to the "new rules". Therefore, the operation of the new drainage pump station is cost-effective and time-efficient.

*2.5. Validation of the Model*

The verification of the built SWMM model was done using an actual flood in the Oncheon River basin on 23 July 2020. This rainfall data was the most recent occurrence with damage information obtainable and the causes of the damage clear. Other damage cases were investigated, but there was no data available for verification.

On 23 July 2020, a flood occurred near Location 2, as shown in Figure 7. The river level exceeded the design flood level which caused the flood to occur. At that time, drainage pump stations continued to pump the water to the Oncheon River, which caused the water level of the river to rise rapidly.

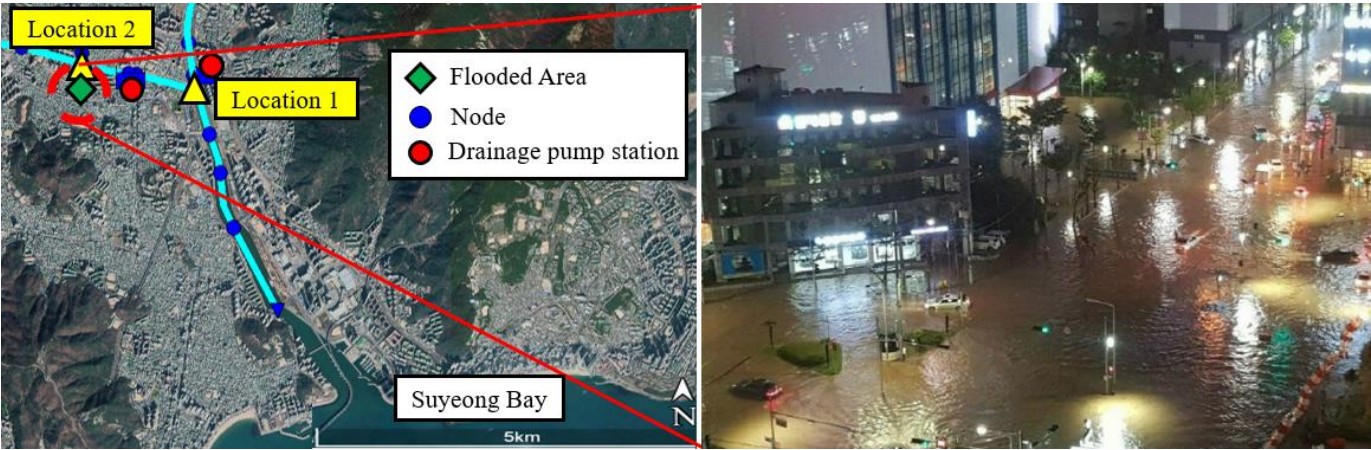

**Figure 7.** Flood Location in 23 July 2020 using SWMM (right photo: provided by Busan Police Agency).

In this study, we simulated rainfall data for 23 July 2020. Simulation results showed that the water level began to rise sharply from 11 p.m. And at 11:30 p.m., Location 2 was already above the design flood level, making drainage impossible (Figure 8). Because the actual flood photo in Figure 7 was taken at 11:30 p.m., the SWMM model we built in this study was simulated based on reality.

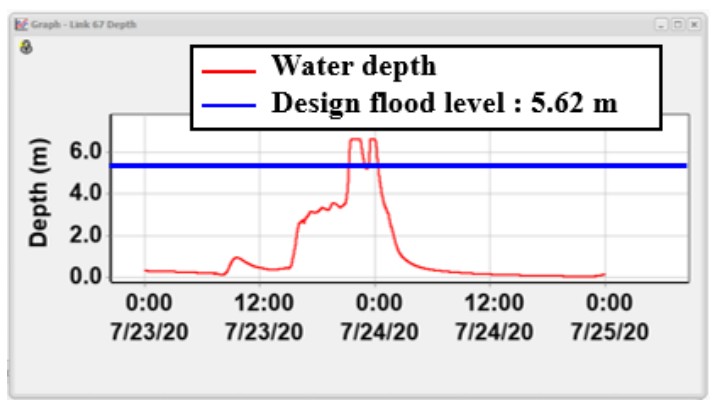

**Figure 8.** Station 2 simulation results for SWMM Model Validation.

## 3. Results

*3.1. Effectiveness of New Methods*

In this study, the design flood levels of Locations 1 and 2 were applied to verify the effectiveness of the operation method of the new drainage pump station. Stations 3, 4, and 5 were stationed at Location 1, and Stations 1 and 2 were sited at Location 2. Figures 9 and 10 show the flood reduction graphs of Locations 1 and 2.

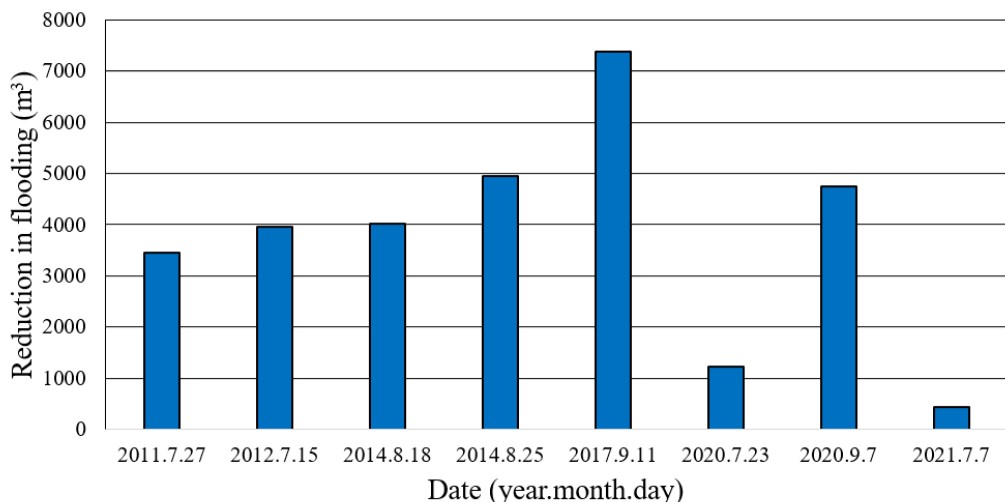

**Figure 9.** Flood reduction at Location 1.

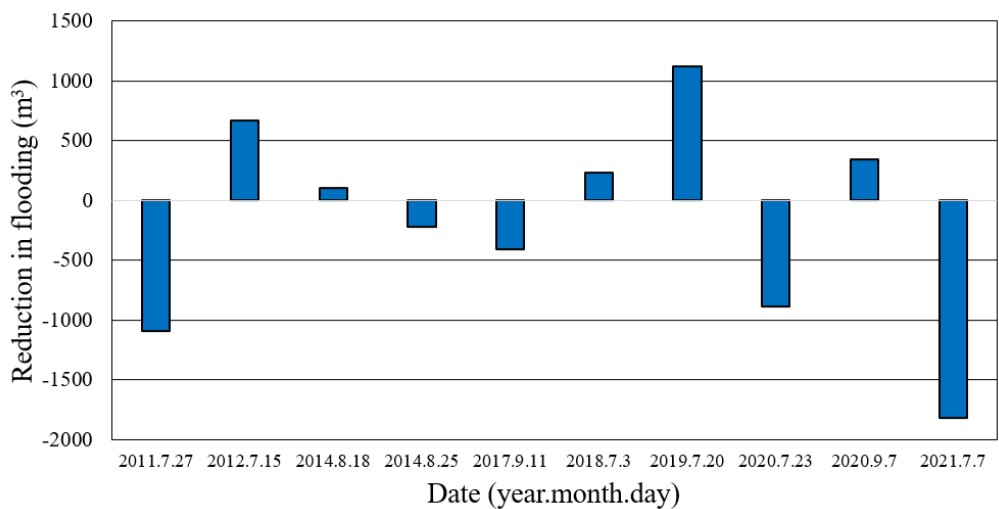

**Figure 10.** Flood reduction at Location 2.

At Location 1 (Figure 9), flooding decreased by an average of 3018.2 m³. Ten rainfall data points were simulated, but no flooding occurred on "2018.07.03" and "2019.07.20", so it was excluded from the graph.

After the 10 rainfall datasets were simulated, only five datasets exhibited reduced flooding at Location 2 (Figure 10), with an average of 495.2 m³. However, on the five data sets, flooding increased. Before explaining the flooding increases, it's worth noting that Location 2 has four problems that make it vulnerable to flooding:

1.  The impermeable area of the Oncheon River is approximately 49%, and most of the area surrounds the stream; hence, rainwater was directly discharged into the stream.
2.  Morphologically, Location 1 is a trapezoidal form, while Location 2 is a very narrow riverbed. This is a very vulnerable form of Location 2 (Figure 11).
3.  Location 2 has a low bed slope (Figure 12).
4.  Because the Oncheon River joins the Suyeong River, the discharge of the Oncheon River is delayed when the water level of the Suyeong River rises.

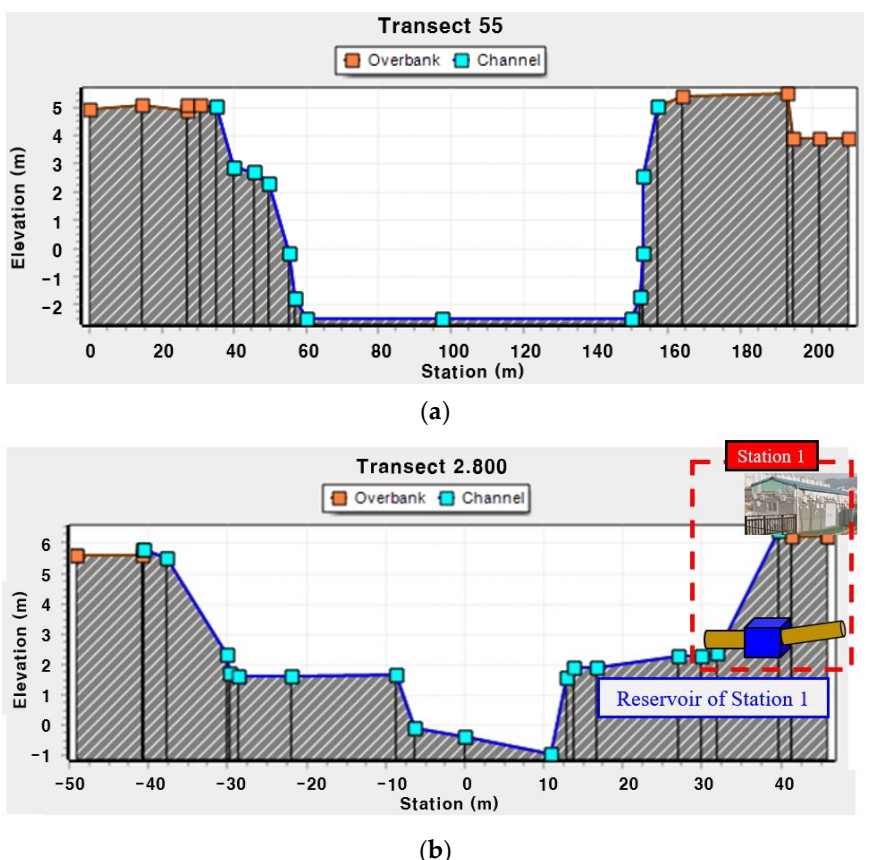

**Figure 11.** Cross-sectional view: (**a**) Location 1; (**b**) Location 2.

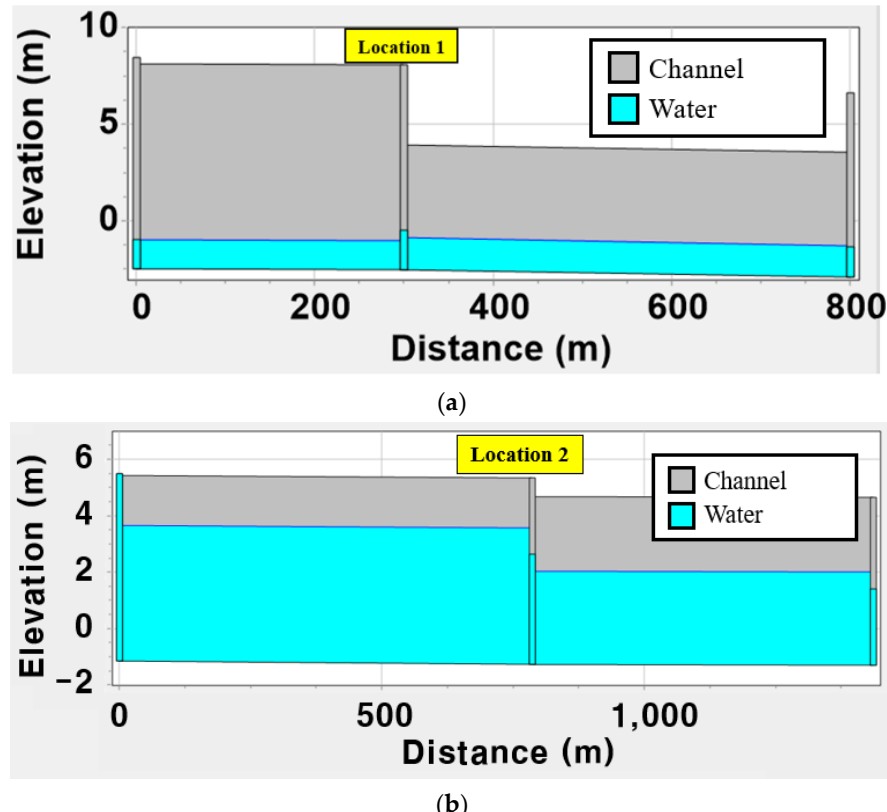

**Figure 12.** Longitudinal sectional view: (**a**) Location 1; (**b**) Location 2.

Therefore, Location 2 has structural problems that are vulnerable to flooding and the reasons for the increased flooding, as shown in Figure 10, are as follows:

1.  The outlet for station 1 is discharged from the walkway located in the middle, not at the top of the channel, as shown in Figure 11b.
2.  The SWMM provided by the U.S. Environmental Protection Agency is a one-dimensional analysis model.
3.  The SWMM 'control' function does not support complex commands.

For three reasons, the outlet of Station 1 is located in the middle of the channel, so the higher the river level, the less drainage there is due to the water pressure (the pump of Station 1 is 120 horsepower). However, since SWMM provides one-dimensional analysis, it drains to the same output even if the water level of the river rises. Therefore, more complex commands are needed to create a situation where drainage is impossible due to the effects of the water pressure. Therefore, in Figure 10, some data with increased flooding can be interpreted as an increase in distributed inland flood data rather than river flooding.

### 3.2. Addressing Structural Problems of Location 2

The flooding of Location 2 increased because the Oncheon River has structural problems that make the river vulnerable to flooding. Therefore, the structural problems of the Oncheon River need to be addressed to apply the proposed method. Virtual storage installations near Location 2 were simulated to compensate for the structural problems of the Oncheon River (Figure 13). The virtual storage was installed underground in the park 190 m from Location 2 and had a small storage area of 45 m² and a depth of 5 m. The pipeline was coded to close when the virtual storage level exceeded 4 m to prevent the flooding of the virtual storage.

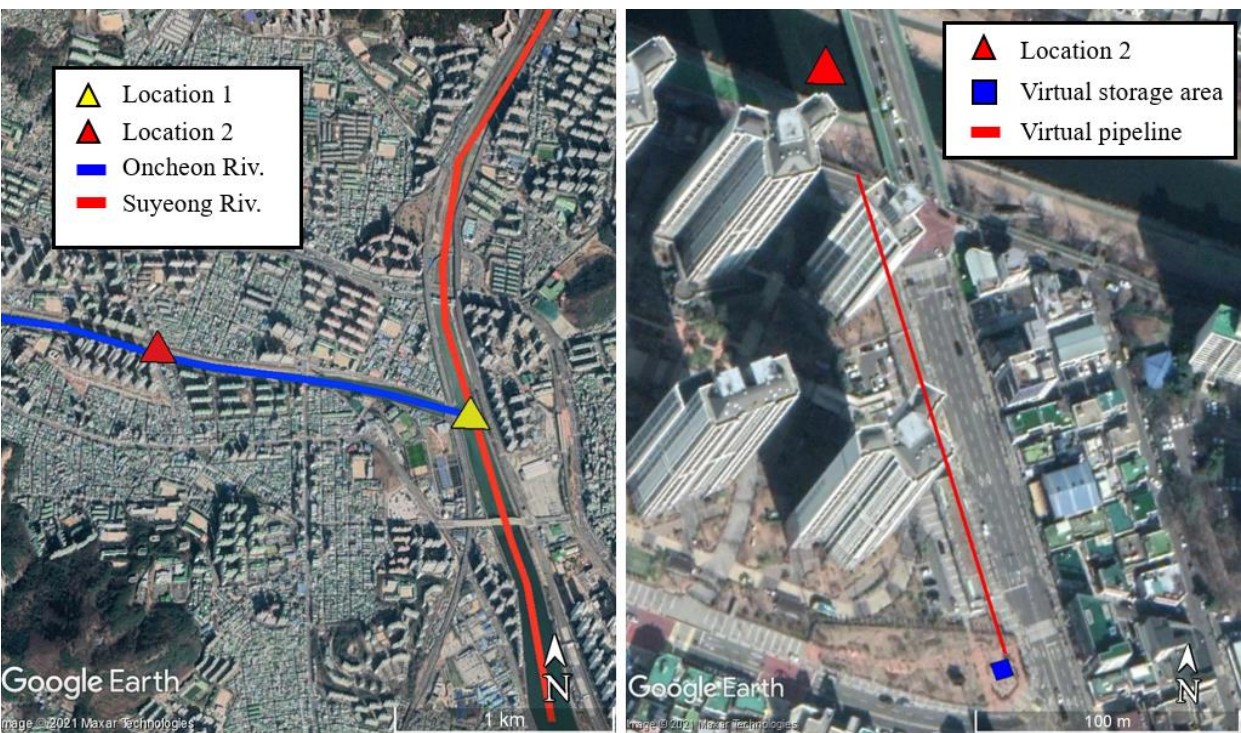

**Figure 13.** Location of virtual storage area.

After virtual storage was installed to solve the structural problems of Location 2, the flood reduction graph was plotted, as shown in Figure 14.

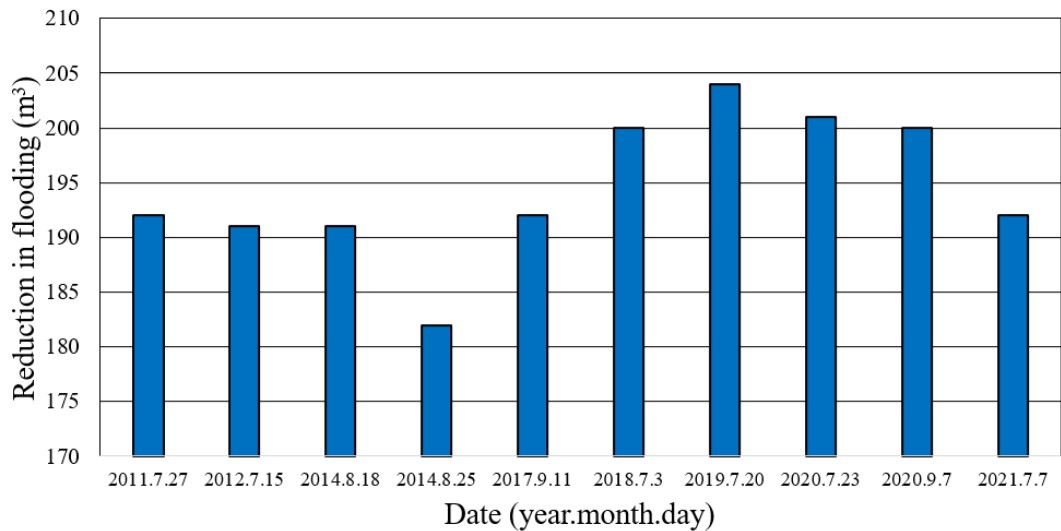

**Figure 14.** Flood reduction at Location 2 after structural problem resolution.

A comparison of Figures 10 and 14 indicates that the Onchen River exhibited a structural problem and was vulnerable to flooding. Supplementing it resulted in sufficient flood reduction through the new drainage pump station operation (Figure 13). Table 4 lists the flooding savings derived from the simulation results.

**Table 4.** Flood reduction values.

| Date | Apply New Operation Method Only | New Operation Method after Structural Troubleshooting | |
|---|---|---|---|
| | Location 1 | Location 2 | |
| | Flood Reduction (m³) | | |
| 2011.07.27 | 3452 | −1093 | 192 |
| 2012.07.15 | 3962 | 668 | 191 |
| 2014.08.18 | 4024 | 107 | 191 |
| 2014.08.25 | 4957 | −217 | 182 |
| 2017.09.11 | 7378 | −412 | 192 |
| 2018.07.03 | none | 232 | 200 |
| 2019.07.20 | none | 1121 | 204 |
| 2020.07.23 | 1222 | −886 | 201 |
| 2020.09.07 | 4750 | 348 | 200 |
| 2021.07.07 | 437 | −1817 | 192 |

## 4. Conclusions

In order to reduce the frequent flooding of rivers due to drainage pump stations, this study proposed a new drainage pump station operation method and simulated it with SWMM. The results were analyzed and proven to be effective due to the reduction of flooding in the river. This is because the goal is to reduce river flooding. The goal is not to reduce inland flooding in the city. Therefore, river flooding has occurred for all of the rainfall data used. And both Location 1 and 2 also caused river flooding in SWMM simulations. When a river is flooded, the river level is at the maximum. Therefore, the analysis was conducted with a reduction amount without analyzing the result through the river level.

Previously, drainage pump stations were operated when the water level of the water pipes increased, but the increase in the river level is not considered when operating the pump. Therefore, if the river level exceeds the design flood level, the drainage pump station stops functioning. This is because the drain at the pump station is usually buried in the embankment, so when the water level rises, it is not properly drained due to the water pressure. If the river level exceeds the design flood level, the flow rate of the river can be easily discharged if the drainage pump station with the running water is stopped. This is because the drainage pump station stops in advance before the river level exceeds the design flood volume, the water from the river drains into the sea, and the drainage pump station operates again. This method is more economical, time-saving, and efficient than building a new drainage pump station.

This study was conducted using the SWMM. SWMM was used for intuitive analysis because it can be applied to flooded basins and used to determine the volume of flooded water. In addition, detailed information on catchment and river information and various facilities in the catchment area can be inputted. SWMM also provides tools for self-coding, which simulated similar to real-world situations.

Ten rainfall data points with precipitation close to 200 mm from the daily rainfall data from 2011 to 2021 were used for the analysis. The analysis results showed that Location 1 had an average reduced amount of 3018.2 $m^3$. This result means that the disaster prevention policy needs to be changed from 'a method to consider only inside water level' to 'a method to consider river level and inside water level'. However, as the investigation progressed, structural problems with hot springs were discovered. Location 2, where the flood reduction of the Oncheon River was analyzed, did not show reduced flooding during some rainfalls, even when the proposed operation method was applied. This was because of the impermeable area, river width, river slope, and embankment level of the Oncheon River. Addressing this problem using a small virtual storage tank resulted in an average flooding reduction of 194.5 $m^3$ at Location 2.

The proposed operation method can effectively reduce river flooding more than the current operation method in areas susceptible to torrential rains caused by climate change. Torrential rains often cause flooding in areas where the river flow does not rapidly drain because it is difficult to predict, and heavy rains fall within a short period. Therefore, as the drainage pump station is constructed, the stream water level rises rapidly, and flooding can occur in all watersheds draining into the stream.

It is necessary to supplement them using a rainwater storage tank to apply the proposed operation method to the Oncheon River. However, the Busan Metropolitan Government can solve the structural problem of the Oncheon River because it periodically builds rainwater storage tanks. In addition, the proposed method requires using the already constructed drainage pump stations, which are applicable to Busan and other cities vulnerable to flooding because it is located at the river downstream. However, the proposed method can only be applied to a basin where a drainage pump station is operated and requires a reservoir for water storage when the drain pump station stops functioning. Therefore, the disadvantage is that it cannot be applied to all watersheds. However, the proposed method is economical and effective for reducing flooding in other regions.

This model can also be considered in combination with other research. In this paper, the geographic complexity of rivers (such as bridges) can be considered and the extent of flooding and depth of flooding can be supplemented using GIS [25–27]. It is then possible to predict the optimal spatial range affected by flooding and to appropriately specify the river water level criteria for the drainage pump stations proposed in this study. Furthermore, flood reduction can be expected at various points in the city, given the additional consideration of the LID [28].

We hope that many cities where drainage pump stations are operated will improve areas vulnerable to damage caused by abnormal weather conditions, and that this research will be further developed to minimize damage in other cities.

**Author Contributions:** Resources, Supervision, Writing—original draft, Y.-M.C.; Investigation, J.-G.K., S.-H.P., T.-H.C.; Conceptualization, Methodology, Software, Writing—review & editing Y.-W.C. All authors have read and agreed to the published version of the manuscript.

**Funding:** This work was supported by BK21 FOUR Program by Pusan National University Research Grant, 2021.

**Institutional Review Board Statement:** Not applicable.

**Informed Consent Statement:** Not applicable.

**Data Availability Statement:** Not applicable.

**Conflicts of Interest:** The authors declare no conflict of interest.

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
