# Peer review of "Method for Operating Drainage Pump Stations Considering Downstream Water Level and Reduction in Urban River Flooding"

_water, doi:10.3390/w13192741_

Round 1
Reviewer 1 Report
- The article is not clear, please explain whether the pumping station in this article refers to the drainage pumping station along the river or the drainage pumping station along the river pipeline network? If flood discharge is along the river, and waterlogging is removed along the river, which should be the latter from the case study of the article, then there is no reduction in flood risk described in the article.
- Line17-18:The article is to reduce flood risk by optimizing the control method of pumping station. Please explain how much flood flow, peak water level, and whether the river water level is safe by this method.
- Line 119-121:Please explain the SWMM model parameters and model calibration process and results.
- Line 123:Why does the article only consider pumping stations and not gates? Is it because the pumping station drains quickly, or the downstream is always at a high tide?
- Line 142:Please explain what parameter of the pump station is Storage area in the table?
- Line 144:Please explain the optimization goals, optimization objects, constraints and solution methods of the algorithm.
- Line 200:The legend and figure in Figure 10 do not match.
- Line 215:Please explain why a reduction in flood volume proves the effectiveness of the method? Shouldn't urban rivers pay more attention to water level and flow? Shouldn't urban waterlogging prevention pay more attention to stagnant water and depth of stagnant water?
- Line 217-218:Please indicate the type and function of the pumping station. Why does the pumping station stop when the river water level exceeds the design water level? How can the pumping station prevent floods?
- Line 229: “had an average reduced area of 3018.2 m3” ?
- Line 235:As shown in Figure 4, point 1 is located below point 2 and closer to the Suyong River. Why is point 1 not affected by Suyong, but point 2 which is farther away is affected?
- Line 239:The article only explains that this method can reduce local floods, but does not explain how to reduce flood disasters. It is recommended to check the definition of flood disasters.
- There are inconsistencies, unclear and even errors in the conceptual logic and language description of this article.

Author Response
Thank you very much to the reviewer who reviewed our paper and gave us a good opinion.
We wrote it in word file reflecting the review comments in detail. Please check the attached file.

Reviewer 2 Report
This research proposed a new rule for controlling pumps in urban drainage system, and compared two ruled-based control strategies in reducing the river flooding. The novelty of this control strategy is not really sufficient, but this research has high value in solving the engineering problem for river flooding control in Busan, Korea.
The manuscript is an original contribution and the topic is of interest for the readership of the Water. The presentation is adequate and clear.
The authors can benefit from the comments below to improve their paper. These have to be accomplished before the manuscript acceptance.
- Major comments
(1) A layout of SWMM model constructed for the study area is suggested to be added to Figure 4. And some description of the SWMM model including number of junctions and conduits should also be added. In addition, the calibration and validation of the SWMM model should be presented.
(2) The four reasons listed at Line 177-184 seem to be the reasons why flooding will occur in Location 2 but not the reasons why the flooding will increase by using new rule. Authors should make further explanation of this result.
(3) The baseline of flooding reduction in Figure 11 is the system without virtual storage in new rule or the system without virtual storage in original rules? If the baseline is new rule, the flooding reduction about 200m3 cannot cover the negative results in some cases, for example, “2011.7.27” and “2021.7.7”. The failure of new rule should be analyzed in quantity, especially for scenario “2021.7.7”, which results shows that the new rule makes flooding increase for 1380m3 considering both Location 1 and Location 2.
- Minor comments
(1) Line 129-130:“the arrival time would cause errors”is confusing, please explain.
(2) Line 231-232: “regardless of the river level” is confusing, the author might mean “regardless of the inside water level”.
Author Response

(The authors gave the same response as above.)

Reviewer 3 Report
The study covers a very important from a practical point of view issue related to the prevention of dangerous floods within large cities. However, there are some points in the work that the authors should more clearly explain to the reader.
1. Your results of reducing flood flow in the studied rivers are simulated. If so, how much are they comparable to observed declines? Was there any validation of the proposed method? How accurate is it?
2. The description of the studied rivers is extremely poor. There is no information about their hydrological characteristics, including intra-annual flow distribution, extreme water discharge, etc. What is the structure of the land cover of the two river basins above localizations 1 and 2?
3. The content of Figures 6, 7, 11, and Table 2 is not clear. This is a reduction in relation to what?
4. Figure 8. What does this figure give to understanding the problem of preventing the rivers from overflowing their banks? The capacity of the channel has a more significant influence on this than its morphology.
5. I did understand neither the content nor the purpose of Figure 9. Where is the comparison with the longitudinal profile of Location 1?
6. Why is the influence of precipitation intensity not analyzed in any way? For example, the same amount of precipitation can fall, in one case, for a whole day, and in the other case, in just a few hours. Obviously, the consequences are different.
7. Lines 217-218. “…operated when the water level increased, but the increase in the river level is ignored.” What does it mean?
8. Lines 110-111. “Data with rainfall values close to 200 mm were used”. How effective will your proposed method be if the daily precipitation is more than 300 mm? It is possible that such extreme precipitation has not been observed since 2011. However, they could have been in earlier periods. How prepared is the city's flood protection system for such extreme events?
9. How effective is the work of the city's flood protection system in conditions of a steady accumulation of suspended and bed-load sediments along the river channel? This accumulation can cause the water levels in the rivers to rise and therefore increase the risk of flooding.
Author Response

(The authors gave the same response as above.)

Reviewer 4 Report
In the submitted manuscript the authors develop a method for operating drainage pump stations to control the river level and verify the effectiveness of the proposed method. A stormwater management model is applied to simulate the Suyeong and Oncheon rivers in Busan, Korea. According to the authors, the introduced method is both economical and efficient for reducing urban stream flooding in areas susceptible to severe damage caused by climate change.
The presented work provides some new data and findings which are of interest and relevant. However, the work is poorly presented and major changes are suggested before this manuscript could be considered for possible publication:
- In the Introduction section the addressed topic and the international relevance of the topic should be introduced in more detail. The scientific goal of the work should be formulated more clearly at the end of this section.
- The Results section reads more like a technical report than like a scientific paper. A careful scientific discussion and well described comparisons with other studies conducted in other study areas worldwide are missing here. These components must be added in this section.
- In the Conclusions section wider and internationally relevant implications of the presented findings must be presented and highlighted more clearly.
- Connected to the previous points: The list of references should be expanded by adding more relevant published works on the topic. The results of these published works should be considered when carefully discussing the own findings.
Figure 1: Please add geographical coordinates here.
Author Response

(The authors gave the same response as above.)

Round 2
Reviewer 1 Report
1. Optimize the structure, logic and expression of the article.
2. Check English usage to improve the accuracy of English expression.
Author Response

(The authors gave the same response as above.)

Reviewer 3 Report
1. Line 17. "Rainfall data used 10 data ...". Please rewrite.
2. Lines 18-19. "The water level of the river was the highest in most simulations." Which river? Suyeong River? Oncheon River?
3. Lines 50-51. Figure 1 shows a satellite map of the lower Suyeong River basin and the entire Oncheon River basin.
4. Caption of Figure 1. Are these the coordinates of what point on the map?
5. The title of Table 1 is incorrect. Better to write "Land-use/-cover structure in the studied river basins". Ratio? Maybe "Share"? Sortation? Maybe "River basin"? Have you assessed the land-use/-cover structure in these river basins concerning the rivers' mouths? If yes, this point should be somehow indicated in the table (or its footnote). Recalculate the data because the SUM must be equal to its components.
6. The headings of the columns in Table 2 are not clear. Only Parameters and nothing else?
7. Lines 157-158. " ... the average tidal curve of Busan is entered and is shown in Figure 4. " Please rewrite. Moreover, why Busan?
8. Figure 12. What do the blue and gray fields mean? There is no legend.
9. The English language of the manuscript needs improvement.
Author Response

(The authors gave the same response as above.)

Reviewer 4 Report
Thank you for addressing my comments.
Author Response
Thanks to your review opinion, our paper could be completed. Your comments have solved the problems of our paper and strengthened international appropriateness.
Thank you so much.